# BTH Treatment Delays the Senescence of Postharvest Pitaya Fruit in Relation to Enhancing Antioxidant System and Phenylpropanoid Pathway

**DOI:** 10.3390/foods10040846

**Published:** 2021-04-13

**Authors:** Xiaochun Ding, Xiaoyang Zhu, Wang Zheng, Fengjun Li, Shuangling Xiao, Xuewu Duan

**Affiliations:** 1South China Botanical Garden, Chinese Academy of Sciences, Guangzhou 510650, China; dingxc111@scbg.ac.cn (X.D.); lifengjun@scbg.ac.cn (F.L.); 2State Key Laboratory for Conservation and Utilization of Subtropical Agro-Bioresources/Guangdong Provincial Key Laboratory for Postharvest Science and Technology of Fruit and Vegetables, College of Horticulture, South China Agricultural University, Guangzhou 510642, China; xiaoyang_zhu@scau.edu.cn (X.Z.); wzeng123456@outlook.com (W.Z.); xiaosl@jxau.edu.cn (S.X.); 3College of Advanced Agricultural Sciences, University of Chinese Academy of Sciences, Beijing 100049, China

**Keywords:** pitaya, BTH, ROS metabolism, phenylpropanoid pathway

## Abstract

The plant resistance elicitor Benzo (1,2,3)-thiadiazole-7-carbothioic acid S-methyl ester (BTH) can enhance disease resistance of harvested fruit. Nonetheless, it is still unknown whether BTH plays a role in regulating fruit senescence. In this study, exogenous BTH treatment efficiently delayed the senescence of postharvest pitaya fruit with lower lipid peroxidation level. Furthermore, BTH-treated fruit exhibited lower hydrogen peroxide (H_2_O_2_) content, higher contents of reduced ascorbic acid (AsA) and reduced glutathione (GSH) levels and higher ratios of reduced to oxidized glutathione (GSH/GSSG) and ascorbic acid (AsA/DHA), as well as higher activities of ROS scavenging enzymes, including superoxide dismutase (SOD), catalase (CAT), ascorbate peroxidase (APX), peroxidase (POD) and glutathione reductase (GR) in comparison with control fruit. Moreover, BTH treatment enhanced the activities of phenylpropanoid pathway-related enzymes, including cinnamate-4-hydroxylase (C4H), phenylalanine ammonia-lyase (PAL) and 4-coumarate/coenzyme A ligase (4CL) and the levels of phenolics, flavonoids and lignin. In addition, BTH treatment upregulated the expression of *HuSOD1/3/4*, *HuCAT2*, *HuAPX1/2* and *HuPOD1/2/4* genes. These results suggested that application of BTH delayed the senescence of harvested pitaya fruit in relation to enhanced antioxidant system and phenylpropanoid pathway.

## 1. Introduction

Pitaya (*Hylocereus undatus*) is a tropical and subtropical non-climactic fruit, rich in vitamins, amino acids, total phenols and betaine. Pitaya is popular among consumers for its attractive color and high-quality taste. However, the fruit is prone to water loss, wilting, rot and rapid senescence, resulting in a short storage life. Till now, techniques are developed to maintain the quality of harvested pitaya, including chemical treatments (BABA, methyl jasmonate, nitric oxide) [1,2], heat shock [3] and low temperature and bagging [4].

Benzo (1,2,3)-thiadiazole-7-carbothioic acid S-methyl ester (BTH) is a structural analog of salicylic acid. BTH exerts no direct bacteriostatic effect on pathogens, but induces plants to acquire systemic immune resistance [5,6]. It has been shown that BTH enhances plant disease resistance and improves the quality of horticultural products such as muskmelon [7,8], strawberry [9], apple [10] and mango [11]. BTH also delays fruit ripening in climacteric muskmelon and banana [12,13]. However, it is still unknown whether BTH regulate the senescence of non-climacteric fruit.

ROS are generated with aerobic respiration, which are implicated in modulating diverse important physiological process as signaling molecules. However, when subjected to environmental stress or during senescence, excessive ROS might accumulate, resulting in oxidative damage of macromolecules, which, in turn accelerate stresses or senescence [14,15]. ROS level is regulated by enzymatic antioxidant system and low molecular mass antioxidants [16,17,18,19,20]. Most fruits have vigorous respiration after harvest, accompanied by ROS accumulation. It has been shown that ROS accumulation and oxidative damage to proteins play roles in fruit senescence [21,22,23]. Previous studies have demonstrated that the postharvest inducer enhances the antioxidant capacity of muskmelon [8], pear [24], blueberry [25] and kiwi fruit [20]. Therefore, maintenance of redox status is crucial for delaying fruit senescence.

The phenylpropanoid pathway plays key roles in the synthesis of phenolic compounds, flavonoids, lignins and other secondary metabolites [26,27]. In the pathway, 4-coumarate-CoA ligase (4CL) and phenylalanine ammonia-lyase (PAL) account for the critical enzymes. Of the synthesized secondary metabolites, some are antioxidants and antibiotic substances, which are important for maintenance of fruit quality and resistance to fungal infection [28]. It has been shown that some exogenous elicitors treatments enhance resistance against fungi and maintains fruit quality by activation of the phenylopropanoid pathway [8,29].

The present work focused on evaluating the role of BTH in regulating the senescence in harvested pitaya fruit and further reveal the underlying mechanism in term of ROS metabolism and phenylpropaoid pathway.

## 2. Materials and Methods

### 2.1. Plant Material and Treatments

Pitaya (*H. undatus* cv. Bai shui jing) fruit were collected at 30 d after anthesis. Preliminary experiment showed that, at a concentration range of 10–100 mg L^−1^, applying 50 mg L^−1^ BTH effectively postponed the harvested pitaya fruit senescence. Therefore, in this study, 50 mg L^−1^ BTH was used. Pitaya fruit were immersed in distilled water (containing 0.05% Tween 80) for 15 min as the control group. In the BTH treatment group, pitaya fruit were dipped in 50 mg L^−1^ BTH solutions (containing 0.05% Tween 80) for 15 min. Each group was carried out independently in triplicate. After treatments, the fruit were placed into 0.02 mm thick unsealed polyethylene bag and stored at room temperature (25 ± 1 °C). Ten fruit are packed in one polyethylene bag, and eight bags are included in each group (BTH and control treatment).

During storage, three fruit in each group were sampled every 2 days, followed by immediate liquid nitrogen freezing and preservation under −80 °C for further analysis. Each treatment was replicated three times.

### 2.2. Weight Loss (WL) Rate

The weights of the pitaya fruit were recorded at 2 d intervals for evaluating weight loss. The WL rate was calculated by the following formula: %WL = (original fruit weight − final fruit weight/original fruit weight) × 100%. Six fruits were used for each replicate of each treatment.

### 2.3. Malondialdehyde (MDA) Content

Malondialdehyde (MDA) content in pitaya peel was determined according to the Malondialdehyde Assay Kit (Beyotime, Shanghai, China). MDA content was expressed as μmol kg^−1^ of fresh fruit weight.

### 2.4. H_2_O_2_ Content

The H_2_O_2_ content was measured according to the method of Gill et al. [30]. The H_2_O_2_ content was monitored at 410 nm and displayed in the manner of mmol kg^−1^ of fresh fruit weight.

### 2.5. Reduced Ascorbate (AsA) and Dehydroascorbate (DHA), Reduced Glutathione (GSH) and Oxidized Glutathione (GSSG) Contents

AsA and GSH contents were determined as the method described by Wei et al. [31]. DHA and GSSG contents were determined using DHA and GSSG Detection Assay Kit (Beyotine, Shanghai, China). AsA and DHA contents were expressed as mg kg^−1^ fresh fruit weight. GSH and GSSG contents were expressed as mmol kg^−1^ on fruit fresh weight, respectively.

### 2.6. Lignin, Total Phenolic and Flavonoids Contents

Lignin, total phenolic and flavonoids contents were analyzed according to the approach of Li et al. [2].

### 2.7. Enzymatic Activity Assays

CAT, APX and GR activities were measured for the reduction in substrates in the reaction systems at 240 nm, 290 nm and 340 nm, respectively, as the methods described in Ren et al. [32], and displayed in the manner of U/mg protein. Typically, one enzyme activity unit was referred to the enzyme volume needed to cause 0.01 absorbance units change/min. SOD activity was determined by evaluating the inhibition of photochemical reduction of nitro blue tetrazolium by the enzyme at 560 nm as the methods previously described in You et al. [19], and expressed as U mg^−1^ protein. Notably, one enzyme activity unit referred to the enzyme volume required to induce 50% inhibition of the reduction. The activities of POD, PAL, 4CL and C4H were determined by evaluating the changes of substrates or reaction products in the reaction systems at 470 nm, 290 nm, 222 nm and 340 nm, respectively, as the methods described in Liu et al. [8], and expressed as U mg^−1^ protein. Typically, one enzyme activity unit indicated the enzyme volume required to induce 0.01 absorbance change per one min. The Bio-Rad Protein Assay Kit (Bio-Rad, Hercules, CA, USA) was utilized to determine protein concentrations within the enzyme extracts.

### 2.8. RNA Extraction and Real-Time Quantitative PCR

Total RNA was extracted from 1 g peel tissue using the RNeasy Plant Mini Kit (Tiangen, China). Thereafter, 1 μg of extracted total RNA was used for preparing cDNA with Reverse-iT 1ST Strand Synthesis Kit (Tiangen, China) according to the manufacturer’s instructions. Gene sequences were obtained from transcriptome data of pitaya from Gene Expression Omnibus (GEO) database GSE119976.

Real-time quantitative PCR was carried out by adding 2 μL of 10 ng/μL template cDNA, 1 μL of each primer (10 μM), sterile water and 10 μL SYBRGreen qPCR Master Mix to total volume 20 μL using the ABI PRISM 7500 Sequence Detection System (Applied Biosystems). PCR was conducted at the conditions shown below, 95 °C for 10 min; followed by 95 °C for 20 s and 58 °C for 40 s for 38 cycles, with *HuACTIN* being an internal control. Appendix A summarizes the specific primers used in qRT-PCR. Three independent biological replicates were used.

### 2.9. Statistical Analysis

Data were expressed as the mean ± SE of three biological replicates. Differences between control and BTH-treated fruit were determined by ANOVA, followed by Student’s *t* test (* *p* < 0.05; ** *p* < 0.01) using SPSS 19.0. Principal component analysis (PCA) was performed using GraphPad Prism9 (GraphPad Software Inc., San Diego, CA, USA). The data were normalized by Z-score [X_std_ = (Xi − $\bar{X}$)/Sx] to ensure that the variables being analyzed were all on similar measurement scales. X_std_ is the standardized value, Xi is the original value, $\bar{X}$ is the variable’s mean and Sx is the variable’s standard deviation.

## 3. Results

### 3.1. Effect of BTH Treatment on Weight Loss and MDA Content of Harvested Pitaya Fruit

Bract yellowing is an important senescence characteristic of harvested pitaya. The bract of the control fruit and 10 mg L^−1^ BTH-treated fruit turned to yellow after 10 d of storage, whereas those of 50 mg L^−1^ and 100 mg L^−1^ BTH-treated remained green (Figure 1A). In the following experiment, we investigated the effect of 50 mg L^−1^ BTH on the senescence of harvested pitaya in relation to redox status and phenylpropanoid pathway.

Moisture plays a vital part in maintaining pitaya fruit quality, while WL can induce fruit wilting. WL was aggravated in control fruit as the storage time extended, which was as high as 4.2% at 10 d after storage. BTH treatment reduced water loss of harvested pitaya. The water loss rate was 2.1% in the BTH-treated fruit after 10 d of storage (Figure 1B).

MDA is an important index of lipid peroxidation, which is related to oxidative stress or senescence in organisms. Application of BTH significantly alleviated lipid peroxidation of harvested pitaya. After 10 d of storage, the MAD contents were 3.6 and 6.0 and μmol kg^−1^ in BTH-treated and control fruit, respectively (Figure 1C).

### 3.2. H_2_O_2_ Content and Activities of CAT, APX and SOD

The oxidative damage of biological macromolecules caused by reactive oxygen species (ROS) is considered to be a main cause of senescence. H_2_O_2_ content in control fruit rapidly increased at 8 d after harvest. BTH treatment significantly reduced the accumulation of H_2_O_2_ content at the later stage of storage (Figure 2A).

CAT, APX and SOD are important antioxidant enzymes responsible for scavenging H_2_O_2_. As shown in Figure 2B,C, CAT and APX activities slightly increased or was constant in control fruit within the first 8 d of storage and then markedly decreased. BTH treatment induced the increase in CAT and APX activities at the early stage of storage and maintained higher levels of CAT and APX activities at the later stage of storage (Figure 2B,C). SOD activity fluctuated at the later stage of storage, and decreased after 6 days of storage. BTH treatment induced and maintained a higher level of SOD activity (Figure 2D).

### 3.3. Activities of GR and POD, Contents of GSH and AsA and Ratios of GSH/GSSG and AsA and DHA

GR activity increased within the first 6 d of storage, and then markedly decreased. BTH treatment resulted in higher GR activity throughout storage (Figure 3A). POD activity tended to decrease in control fruit during storage. BTH treatment delayed the decrease in POD activity (Figure 3B). GSH content was generally constant in control fruit during storage. BTH treatment resulted in higher GSH level at the later stage of storage (Figure 3C). AsA content tended to increase during storage. BTH treatment maintained higher AsA level (Figure 3D).

Moreover, the ratios of reduced to oxidized glutathione (GSH/GSSG) and ascorbic acid (AsA/DHA) were high in BTH-treated fruit than in control fruit (Figure 3E,F).

### 3.4. Activities of PAL, C4H and 4CL

PAL and C4H activities in control fruit showed a slight increase during storage. BTH induced the increases in PAL and C4H activities within the first 4 d of storage and maintained higher levels of PAL and C4H activities in the late storage period (Figure 4A,B). 4CL activity in control fruit initially increased slightly and then decreased during storage. In contrast, BTH treatment initially induced the increase in 4CL activity, and then the activity was maintained at a high level (Figure 4C).

### 3.5. Total Phenols, Flavonoids and Lignin Contents

Total phenolic content slightly increased within the first 6 d of storage and then decreased in control fruit. Application of BTH promoted the synthesis of total phenolic content. Total phenolic content was higher in BTH-treated fruit than in control fruit throughout storage (Figure 5A). Flavonoid content was constant over the first 6 d of storage, then decreased in control fruit. BTH treatment resulted in increased accumulation of flavonoids during the later stage of storage (Figure 5B). Lignin content was virtually constant in control fruit throughout storage. Application of BTH increased the synthesis of lignin and maintained a higher level of lignin in harvested pitaya throughout storage (Figure 5C).

### 3.6. Expression of HuSODs, HuAPXs, HuCATs and HuPODs

To illuminate the response of harvested pitaya fruit to BTH, we evaluated the expression profiling of ROS metabolism-associated genes, including four *HuSODs*, two *HuCATs,* three *HuAPXs* and three *HuPODs*. BTH treatment upregulated the expression of *HuSOD1/3/4* genes, but downregulated the expression of the *HuSOD2* gene. Except after day-4 and day-6, the expression of *HuCAT1* was not affected by BTH, but the expression of *HuCAT2* was upregulated by BTH. Of the analyzed three *HuAPXs* and three *HuPODs*, the expression of *HuAPX1/2* and *HuPOD1/2/4* were upregulated by BTH in comparison with the control.

## 4. Discussion

BTH is a synthetic plant elicitor that has similar function and structure with SA. There is mounting evidence that BTH effectively induces disease resistance in harvested fruits, such as muskmelons [7,8], strawberries [9], apples [10] and mangos [11]. BTH-induced disease resistance might be related to altered ROS metabolism, activated phenylpropanoid pathway, induced pathogenesis-related protein and accumulation of resistant substances [33]. Recent studies have revealed that BTH also delays fruit ripening in climacteric fruits, such as muskmelon and banana [12,13]. Nonetheless, it remains largely unclear whether BTH is involved in the regulation of the senescence of non-climacteric fruit. Pitaya is a non-climacteric fruit that undergoes senescence after postharvest, resulting in decreased quality [34]. In the present study, BTH pretreatment obviously inhibited turning yellow of bract and water loss of harvested pitaya fruit. Moreover, lipid peroxidation, a senescence-related index, was significantly reduced in harvested pitaya fruit by BTH. Our results indicate that BTH effectively delays the senescence of harvested pitaya, which is possibly related to maintenance of redox status.

ROS are inevitable by-products of aerobic metabolism in organisms, which play important roles as signaling molecules in regulating growth and development, metabolism and response to environmental challenges. In plants, low concentrations of ROS induce the synthesis of plant disease-related proteins and participate in the cross-linking and lignification of cell walls to resist fungal infections [35]. However, excessive ROS accumulation due to imbalance of production and elimination will result in oxidative damage to macromolecules, thereby accelerating senescence [15]. Fruit senescence has been recently proposed to be associated with ROS accumulation and protein oxidative injury [21,22,36]. Qin et al. [22] reported that protein oxidation was intensified with the proceeding of senescence in peach fruit, and low temperature inhibits ROS accumulation, alleviates protein carbonylation and retards fruit senescence whereas H_2_O_2_ treatment results in the opposite effect. Wu et al. [36] found that high oxygen concentration resulted in higher H_2_O_2_ accumulation and accelerated fruit senescence in longan fruit whereas exposure of low oxygen has the opposite effect. In the present study, H_2_O_2_ rapidly accumulated in pitaya fruit during the later storage, in consistence with fruit senescence. BTH treatment resulted in the increased H_2_O_2_ level during the initial storage, which might serve as a signal molecule to induce antioxidant enzyme activities [37], but reduced H_2_O_2_ accumulation at the later stage of storage. It is suggested that the delayed senescence by BTH is related to decreased accumulation of ROS.

A complicated non-enzymatic and enzymatic antioxidant mechanism has evolved in plants to cope with oxidative stress [30,38,39]. Those major antioxidants are APX, SOD and CAT. SOD is responsible for catalyzing superoxide radical dismutation into H_2_O_2_. CAT directly eliminates H_2_O_2_ while APX degrades H_2_O_2_ using AsA as the electron donor [40]. In addition, GR catalyzes the reduction of oxidized glutathione, which facilitates APX to function [40]. Apart from antioxidant enzymes, glutathione, ascorbic acid, tocopherols and phenolic compounds constitute the majority of non-enzymatic antioxidants in plants, which directly eliminate ROS, provide reducing power or maintain redox state of microenvironment [18,41,42]. Numerous studies have shown that upregulated expression of *SOD*, *CAT* and *APX* or increased activities of antioxidant enzymes reduce ROS accumulation and therefore increase resistance to abiotic stress in plants [17,18,19]. BTH-induced disease resistance also might be related to enhanced antioxidant enzyme activities [9,32]. In this study, application of BTH upregulated expression of *HuSOD1/3/4*, *HuCAT2* and *HuAPX1/2*, *HuPOD1/2/4* (Figure 6) and increased activities of SOD, CAT and APX (Figure 2B–D), in consistence with the alleviated accumulation of H_2_O_2_ (Figure 2A). Moreover, the increase in GR activity (Figure 3A), contents of GSH and AsA and ratios of GSH/GSSG and ASA/DHA (Figure 3C–F) in BTH-treated fruit compared with control fruit indicated that the redox state was well maintained by BTH, which are important for APX to eliminate H_2_O_2_. Peroxidase (POD) is a hemoprotein catalyzing the oxidation of a number of substrates, such as phenols, amine compounds and hydrocarbon oxidation products, using hydrogen peroxide as the electron acceptor. POD works with other antioxidant enzymes to eliminate excess ROS [43,44]. Our results showed that BTH treatment up-regulated expression of *HuPOD1/2/4* and increased POD activity in harvested pitaya fruit (Figure 3B and Figure 6). Thus, BTH treatment activated the gene expression and activities of antioxidant enzymes, and well maintained the redox status, which might be responsible for the alleviated H_2_O_2_ accumulation and the delayed senescence in harvested pitaya fruit (Figure 7). To further predict the key redox parameters of enzyme activity, metabolite content and related gene expression in the BTH-treated pitaya fruit, we performed a multivariate PCA analysis between BTH-treated and control fruit (Appendix A). The results showed that PC1 and PC2 together accounted for 6.34% of the cumulative proportion of variance and 22 redox variables were significantly separated between the BTH-treated and control group (Appendix A). In the loadings plot for our data, the activity of CAT, GR, the content of H_2_O_2_, AsA and GSH, the ratio of AsA/DHA and GSH/GSSG, the mRNA transcripts levels of *HuSOD1, HuSOD2, HuSOD3, HuSOD4, HuCAT2, HuAPX1, HuPOD1, HuPOD2* and *HuPOD4* were strongly correlated (values close to 1 or −1, *p* < 0.05) (Appendix A).

The phenylpropanoid pathway is considered as a rich source of secondary metabolites, such as phenolic compounds, flavonoids, coumarins and lignins in plants. In phenylpropanoid pathway, PAL, 4CL and C4H serve as the critical enzymes, of which PAL and C4H are the first and second rate-limiting enzymes [45,46], whereas 4CL is the key enzyme leading to the branch of phenylpropanoid pathway [26]. Phenols and flavonoids are the end products of phenylpropanoid pathway, and their accumulation are closely associated with plant resistance and senescence [47,48,49,50,51]. Lignin enhances the mechanical strength of plant cell walls and prevents microbial invasion [29], and is involved in regulation of senescence [52,53,54,55]. In the present study, application of BTH promoted PAL, 4CL and C4H activities (Figure 4), in consistence with the increased synthesis of phenols, flavonoids and lignin (Figure 5) in pitaya fruit. Li and colleagues also came to consistent results in red pitaya [1], who found that beta-aminobutyric acid treatment induces resistance against rot caused by *Gilbertella persicaria* via activating phenylpropanoid. It appears that the increased synthesis of phenols, flavonoids and lignin in BTH-treated pitaya fruit is important to maintain redox state and increase resistance, thereby delaying postharvest pitaya fruit senescence (Figure 7).

## 5. Conclusions

BTH treatment effectively delayed postharvest pitaya fruit senescence. The delayed senescence was related to enhanced antioxidant system and phenylpropanoid pathway. BTH treatment upregulated expression of antioxidant enzyme genes, increased activities of antioxidant enzymes and regenerative capacity of glutathione and ascorbic acid. Moreover, BTH treatment also induced PAL, C4H and 4CL activities, resulting in the increased synthesis of phenols, flavonoids and lignin. Nevertheless, the investigation of the mechanism of BTH to delay pitaya fruit senescence is still needed to be further studied at the molecular level.

## Figures and Tables

**Figure 1 foods-10-00846-f001:**
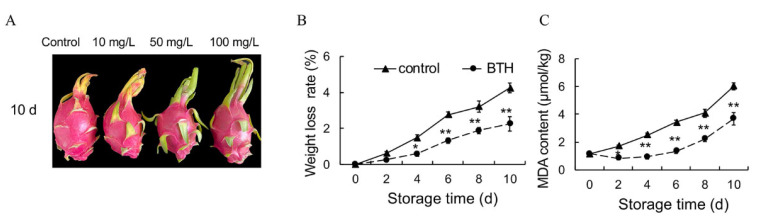
Senescence characteristics of pitaya fruit during storage at 25 °C. (**A**) Visual appearance of pitaya fruit treated with 0 (control), 10, 50 and 100 mg ^−1^ BTH after 10 d of storage. (**B**,**C**) Effect of BTH treatment on weight loss rate (**B**) and malondialdehyde (MDA) content (**C**) in pitaya fruit during storage. The concentration of BTH was 50 mg^−1^. The data are presented as the mean ± SE of three replicates. Asterisks indicate the significant differences between the two groups (Student’s *t* test, * *p* < 0.05; ** *p* < 0.01).

**Figure 2 foods-10-00846-f002:**
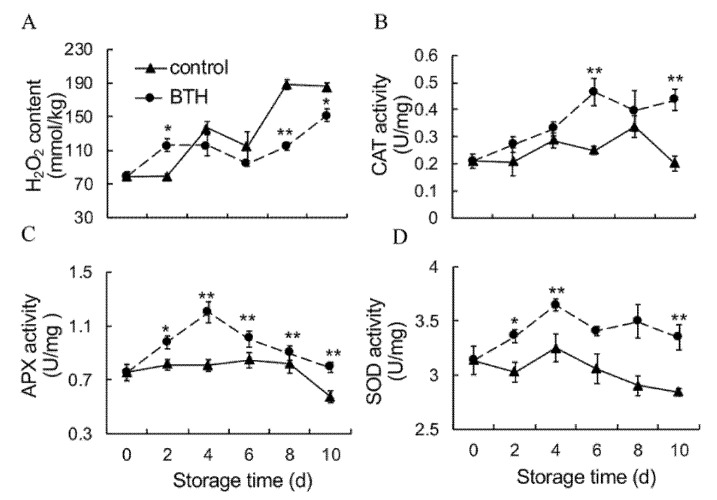
Effect of BTH treatment on H_2_O_2_ content (**A**), activities of CAT (**B**), APX (**C**) and SOD (**D**) in pitaya fruit during storage. H_2_O_2_, hydrogen peroxide; CAT, catalase; APX, ascorbate peroxidase; SOD, superoxide dismutase. Asterisks indicate the significant differences between the two groups (Student’s *t* test, * *p* < 0.05; ** *p* < 0.01).

**Figure 3 foods-10-00846-f003:**
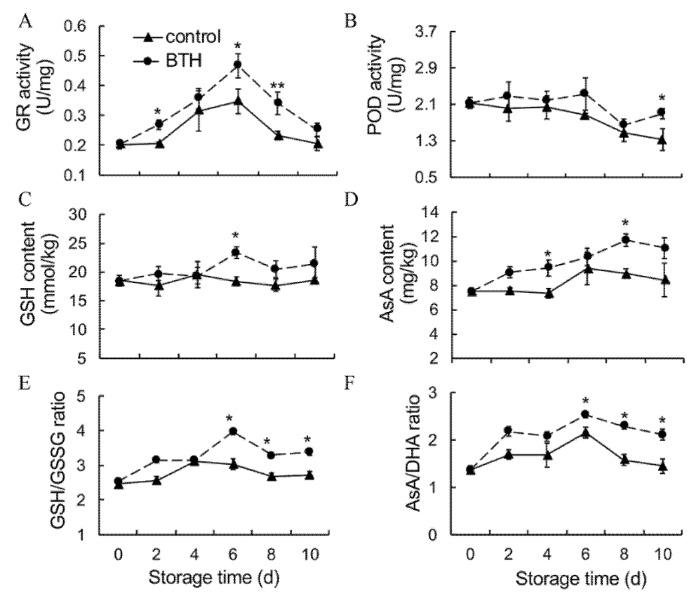
Effect of BTH treatment on activities of GR (**A**) and POD (**B**), contents of GSH (**C**) and AsA (**D**) and ratios of GSH/GSSG (**E**) and ASA/DHA (**F**) in pitaya fruit during storage. GR, glutathione reductase; POD, peroxidase; GSH, glutathione; AsA, ascorbic acid; GSSG, oxidized glutathione; DHA, oxidized ascorbic acid. Asterisks indicate the significant differences between the two groups (Student’s *t* test, * *p* < 0.05; ** *p* < 0.01).

**Figure 4 foods-10-00846-f004:**
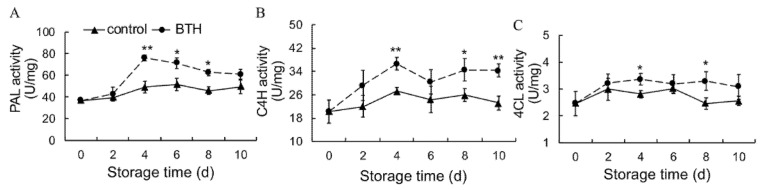
Effect of BTH treatment on activities of PAL (**A**), C4H (**B**) and 4CL (**C**) in pitaya fruit during storage. PAL, phenylalanine ammonia-lyase; C4H, cinnamate-4-hydroxylase; 4CL, 4-coumarate/coenzyme A ligase. Asterisks indicate the significant differences between the two groups (Student’s *t* test, * *p* < 0.05; ** *p* < 0.01).

**Figure 5 foods-10-00846-f005:**
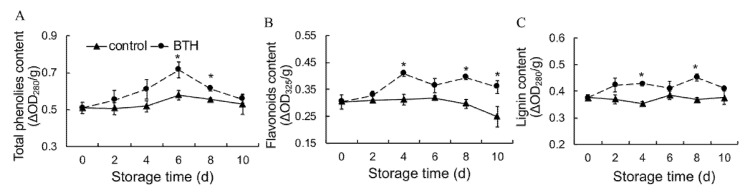
Effect of BTH treatment on contents of total phenolics (**A**), flavonoids (**B**) and lignin (**C**) in pitaya fruit during storage. Asterisks indicate the significant differences between the two groups (Student’s *t* test, * *p* < 0.05).

**Figure 6 foods-10-00846-f006:**
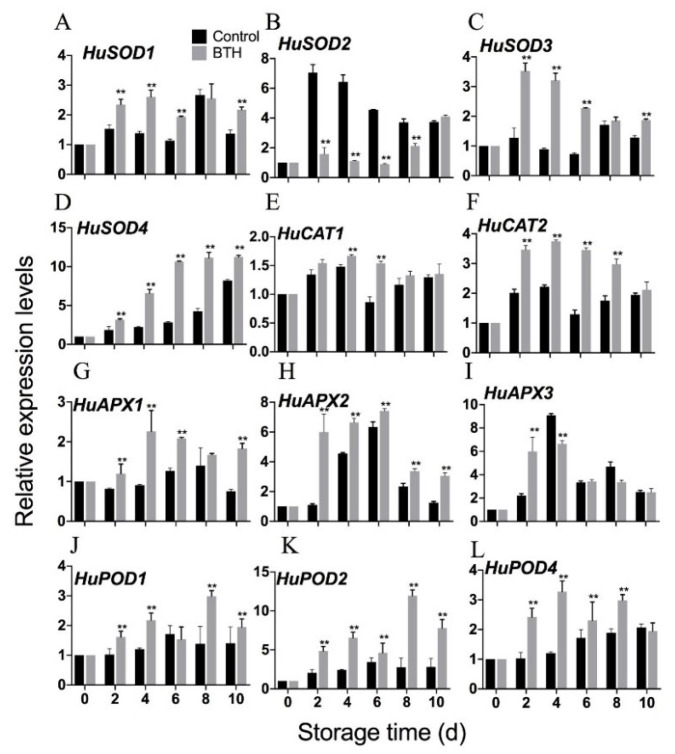
Expression levels of *HuSODs* (**A**–**D**), *HuCATs* (**E**–**F**), *HuAPXs* (**G**–**I**) and *HuPODs* (**J**–**L**) in pitaya fruit treated with or without BTH during storage. *HuACT* was used as the reference gene. Gene expression of the sample from 0 d was set as 1. Asterisks indicate the significant differences between the two groups (Student’s *t* test, ** *p* < 0.01).

**Figure 7 foods-10-00846-f007:**
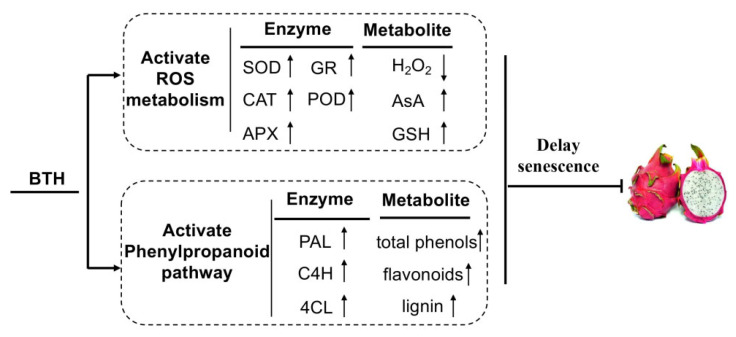
A proposed model of the involvement of BTH in the regulation of fruit senescence in pitaya. Arrow up means promotion, arrow down indicates suppression.

## Data Availability

The data presented in this study are available on request from the corresponding author.

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
