# Peer review of "BTH Treatment Delays the Senescence of Postharvest Pitaya Fruit in Relation to Enhancing Antioxidant System and Phenylpropanoid Pathway"

_foods, 2021, doi:10.3390/foods10040846_

Round 1

Reviewer 1 Report

This study examines the effect of the salicylate analogue BTH on the redox metabolism of senescing pitaya fruit.

While it is an excellent piece of work increasing our understanding of redox metabolism in a crop fruit, there are significant concerns that need to be addressed in a revised version of this manuscript.

First, there is no bibliography list, which makes it very difficult to assess the relevance of the citation, or if some seminal work is missing (e.g. towards the short introduction). Furthermore, BTH treatment is not detailed in the Methods section (spray, soil drenching, etc ?). What tissue/organ is being treated?

Second, while AsA and GSH contents are informative, it would be best to determine the redox state of these redox buffers, expressed as reduced over oxidised pools, for instance.

Third, this work is very descriptive, and perhaps a multivariate analysis (PCA, HCA, etc.) of the different redox parameters could integrate better these results (metabolite vs enzyme vs transcripts)

Reviewer 2 Report

The current manuscript address an interesting problem, delay the senescence of pitaya, extending the postharvest life and prolonging fruit quality. Authors investigated the effects of Benzo (1, 2, 3)-thiadiazole-7-carbothioic acid S-methyl ester (BTH) on the rate of water loss, and several biochemical and molecular makers related to antioxidant capacity.  The objective of the study is focused on the BHT-induced antioxidant capacity and authors analyzed changes in enzyme activity and the expression of the antioxidant enzymatic system. Moreover, changes in phenolics, flavonoids and lignin were also analyzed. I think that the manuscript is reports interesting data, with promising results, is well performed, the experimental work is designed and performed.

There are however, few points that I think should be clarified:

  • One major argument is about interpretation f PAL on the system. From the results, it is clear an increase in PAL activity by BHT, however, changes in gene expression did not correlated with enzyme activity. Authors did not provide any explanation in the discussion or at least a discussion of the situation.
  • Transcripts of PAL1 increase at day 4 and 10, and those of PAL2 at days 4, 6 and 10. I think, then that there not evidence enough to support an increase or a decrease, since both situations are compatible with the data. Such statement are indicate in the abstract and I do not is relevant and consistent with the whole data.
  • It seems evident the increase in PAL, C4H (Fig. 4) but are the data of 4CL significant?. I do not think that such differences are enough to formulate a clear statement for this enzyme activity.
  • From the above arguments, I think that authors should revised the abstract accordingly and I suggest to reconsider the title, since it seems that the pneylpropanoid pathway is enhanced in a similar fashion that antioxidant system, that is not the case.

Round 2

Reviewer 1 Report

I thank the authors for addressing the comments, but some of them still require attention.

  • The first occurrence of ref 14 could be complemented by a general review on redox metabolism in fruit (eg Decros et al, 2019 Frontiers in Plant Science). This will give a broader context to this work.
  • PCA presented in Supp Fig 1 lacks important information : what normalisation has been used, how many variables are considered, what are the variances explained by PC1 and PC2. Besides, the figure itself has a very low quality image. This should be improved. Panel B is very confusing, is this a loading plots ? This should also be revised.
